# Proposing and Validating the Diagnosis Scale for Internet Gaming Disorder in Taiwanese ADHD Adolescents: Likert Scale Method Based on the DSM-5

**DOI:** 10.3390/ijerph18041492

**Published:** 2021-02-04

**Authors:** Yue-Cune Chang, Ruu-Fen Tzang

**Affiliations:** 1Department of Mathematics, Tamkang University, New Taipei 251, Taiwan; ycchang414@gmail.com; 2Department of Psychiatry, Mackay Memorial Hospital, Taipei 104, Taiwan; 3Department of Childhood Care and Education, Mackay Junior College of Medicine, Nursing, and Management, Taipei 252, Taiwan; 4Department of Medicine, Mackay Medical College, Taipei 252, Taiwan

**Keywords:** Taiwan IGD-SF-L, DSM-5, ADHD, ROC curve, area under the curve (AUC)

## Abstract

The paper aims to adjust the Taiwanese version of Internet gaming disorder-short form Likert scale with Likert (IGD-SF-T-L) based on the Diagnostic and Statistical Manual of Mental Disorders (DSM-5) criteria to a Likert scale model and test its psychometric property among children and adolescents with Attention Deficit Hyperactivity Disorder (ADHD). Confirmatory factor analysis (CFA) was conducted for validity and the Cronbach’s α for reliability of IGD-SF-T-L. The ROC (receiver operating curves) was used to propose the cut-off point for assessing the instrument’s psychometric properties and its corresponding indices for the diagnostic accuracy. In total, 102 children and adolescents with ADHD were recruited. The construct validity of IGD-SF-T by CFA was model well fitted with excellent reliability (Cronbach’s α = 0.918). The ROC using the Chen’s CIAS > 56 as the state variable for IGD diagnosis showed the AUC (areas under the curves) was 0.918. The cut-off point proposed for IGD-SF-T-L to indicate a diagnosis of IGD was ≥ 10. The corresponding indices of accuracy: sensitivity, specificity, LR (likelihood ratio) +, LR-, and AUC were 0.893, 0.826, 5.134, 0.130, and 0.859, respectively. The proposed IGD-SF-T-L is an adequate, standardized psychometrical measurement for diagnosing IGD among Taiwanese adolescents with ADHD. More attention should be paid toward recent ADHD youth with Internet gaming disorder and their family.

## 1. Introduction

Nowadays, playing Internet-based games is a novel, normal, enjoyable, and often well-accepted sociocultural practice [1,2]. However, for child and adolescent, the behavior of pathologically excessive and problematic gaming might defer their personality development through children’s long time significant psychological distress and impairment in an individual’s life [3,4]. When youth become victim of Internet gaming disorder (IGD), they may commonly characterize with defective self-strength, dysregulated mood or reward system, problems on decision-making, lack of social skills, and adverse family–child interaction [5]. Accordingly, quite a lot of youth were actually home bounded with gaming addiction without any treatment [6]. Gradually, new juvenile mental health related crisis might be rising due to prevailing untreated problematic gaming on child and adolescent.

In 2018, the World Health Organization has formally included Gaming Disorder in the ICD-11 as an official mental health disorder. In the latest (fifth) edition of the Diagnostic and Statistical Manual of Mental Disorders (DSM-5) of the APA (American Psychiatric Association) [7] has proposed Internet gaming disorder (IGD) as a tentative mental disorder under section III. Indeed, Internet addiction brings tremendous psychological damage to recent youth. Child mental health experts began to pay attention to this adolescent’s gaming disorder problem and suggest child and adolescent psychiatrists all over the world to diagnose and treat IGD earlier. Therefore, there is a special need to study the psychometric properties of measurement tool of IGD according to different population from different country.

The Internet Gaming Disorder Scale-Short-Form (IGDS9-SF) developed by American Psychiatric Association (APA) is quite popular measurement of IGD [8]. The validity of IGD has been confirmed in the Spanish-[9], Portuguese-[10], Slovenian-[11], Italian-[12], English-[13], and Hungarian-[14] speaking samples. In Asian countries, Ko et al. had demonstrated IGD-SF criteria of the DSM-5 had high diagnostic validity by using a semi-structured interview schedule in a Taiwanese sample [15]. In a Hong Kong-based study, the IGD criteria of DSM-5 were translated into a Chinese IGD scale (C-IGDS) that also shown good validity by a self-reporting scale, a telephonic, population-based survey study of adults [16].

The validity of DSM-5’s IGD scale largely comprised adults before [17] only few for adolescents [11]. However, Koronczai et al. pointed out that the study of psychometric properties with good quality should also contain the different clinical samples from different cultures or countries instead of focusing only on adult aged groups [18]. The gaming disorder for child and adolescent population is not only the problem of gaming excessively and frequently. More serious problem is IGD were co-occurring with other mental disorder like attention-deficit/hyperactivity disorder (ADHD), obsessive-compulsive disorder (OCD), anxiety, and depression, somatization, obsession-compulsion, interpersonal sensitivity, hostility, phobic anxiety, paranoid ideation, and psychoticism [19,20]. According to a comprehensive review, there is a close association between gaming disorder and ADHD [21]. Clinically, many untreated hyperactive children have been seen Internet addiction problem at the same time. Both IGD and ADHD is quite commonly seen neurodevelopmental mental disorder among child and adolescent, as prevalence of IGD: 2% [5] vs. ADHD: 5.29% [22]. The chance of comorbidity of these two diseases is ranged from 29% [23] to 83.3% [24]. Clinician even doubt one of the bad consequences of untreated ADHD children is Internet gaming disorder. Therefore, it is important to check the psychometric property of IGD-SF from DSM-5 among different clinical populations like ADHD youth among different countries, like Asian cultures or countries. In addition, a previous validation study of the DSM-5’s IGD criteria used “yes” or “no” response options for each item. Such response model might not assure the accurate responses. For a more accurate response, a Likert scale rather than a “yes” or “no” model might provide more suitable assessment of IGD-SF.

Therefore, this study set out an aim to test the more complete psychometric property of Taiwanese version of the Internet Gaming Disorder Scale with Likert scale (IGD-SF-T-L), including reliability, construct validity, the ROC (receiver operating curves) using the Chinese Internet Addiction Scale as the state variable for diagnosing IGD to propose the cut-off point and its corresponding indices to comparing the diagnostic accuracy for assessing the instrument’s psychometric properties of IGD-SF-T-L from DSM-5 for children and adolescents with ADHD from an Asian country—Taiwan.

## 2. Methods

### 2.1. Participants and Data Collection

A total of 102 children and adolescents (mean age = 11.16 ± 3.35 years, 68.6% boys) were recruited from the outpatient units of the Mackay Memorial Hospital (MMH) in Taipei, Taiwan. The MMH Institutional Review Board (IRB) approved the research protocol. Written informed consent was obtained from each subject as per the IRB guidelines. According to the inclusion criteria, boys and girls with ADHD aged 7–18 years were enrolled. If the patients or their parent(s) or caregiver(s) suspected they had psychotic disease, mental retardation, or other mental conditions that could prevent them from completing the study, they were excluded. After obtaining signed consent from a legal guardian, each participate were interviewed for the following measurements.

### 2.2. Measurements

#### 2.2.1. Chen Internet Addiction Scale for Gaming Disorder

The Chen Internet Addiction Scale (CIA) is a 4-point, self-reported questionnaire comprising 26 questions that assess the five dimensions of Internet use-related problems, namely compulsive use, withdrawal, tolerance, interpersonal and health problems, and time management problems [25]. The scale has good reliability and validity. The internal reliability of the scale and the subscales in the original study ranged from 0.79 to 0.93. Higher CIA scores indicated greater severity of IA. The CIA scale has good diagnostic accuracy (89.6%). The screening cut-off point has high sensitivity (85.6%), and the diagnostic cut-off point has high diagnostic accuracy, as indicated by the correct classification rate of 87.6% of the participants.

#### 2.2.2. Swanson, Nolan, and Pelham, Version IV Questionnaire for ADHD and ODD

The Swanson, Nolan, and Pelham, Version IV questionnaire (SNAP-IV) consists of the following items: inattention, hyperactivity/impulsivity, and oppositional symptoms. These items reflect the core symptoms of ADHD and Oppositional Defiant Disorder (ODD) as defined in the DSM-IV. A study of the psychometric properties of the Chinese version of the SNAP-IV in Taiwan showed that the intraclass correlation coefficients for the three subscales of this scale ranged from 0.59 to 0.72 for the parent form and 0.60 to 0.84 for the teacher form. All subscales of the parent and teacher forms showed excellent internal consistency with Cronbach’s α values of >0.88 [26].

#### 2.2.3. Taiwanese Version of the Internet Gaming Disorder Scale with Likert Scale (IGDS-SF-T-L)

In Taiwan, Ko and colleagues had translated DSM-5’s IGD and found good validity [15]. We transformed the Taiwan IGD-SF criteria from a “yes” or “no” model to a 4-point Likert scale, with scores ranging from 0 (never) to 3 (very often) and total scores ranging from 0 to 27. Higher scores indicated more IGD.

### 2.3. Statistical Analyses

For socio-demographic information of the subjects with and without an Internet gaming disorder, we used the Fisher’s exact test for categorical data and independent *t*-test for continuous data. We investigate construct validity of the IGDS-SF-T by Confirmatory factor analysis (CFA). We used several model fit indices including root-mean-squared error of approximation (RMSEA), and comparative fit index (CFI) to test CFA model for getting best represent of the present dataset. RMSEA is a measure of the average of the residual variance and covariance; good models have RMSEA values that are at or less than 0.08. CFI is an index that fall between 0 and 1, with values greater than 0.90 considered to be indicators of good fitting models. When comparing models, a lower chi-square value indicates a better fit, given an equal number of degrees of freedom.

Diagnostic accuracy associate with the discriminating ability between the target condition and health. This discriminative potential can be quantified by the measures of diagnostic accuracy such as sensitivity and specificity, predictive values, likelihood ratios, the area under the ROC curve [27]. In order to determine levels of accuracy, we calculated receiver operating characteristic (ROC) curves, area under the curve (AUC) of the ROC, sensitivity, specificity, as well as likelihood ratios. The area under the curve (AUC) has a meaningful interpretation for diagnostic indexes of accuracy. The advantage of ROC curve is to determine the optimal cut off values [28]. The diagnostic accuracy (sensitivity, specificity, and positive/negative likelihood ratios) also can be used to clarify the diagnostic concordance between the Taiwanese version of the IGDS and the CIA (56/57). Sensitivity was calculated as the probability of a person with a score of ≥57 on the CIA being diagnosed with IGD according to the IGDS-SF-T. Specificity was calculated as the probability of the Taiwanese version of the IGDS not diagnosing a person with IGD when their CIA score is ≤56. We used the likelihood ratio (LR) for evaluating the diagnostic accuracy instead of using the positive and negative predictive values. The LRs are independent of prevalence and are defined as follows: LR+ = sensitivity/(1 − specificity) and LR− = (1 − sensitivity)/specificity. A test with an LR+ value of >10 or an LR-value of < 0.1 is likely to be “very useful test”, and an LR+ value of 2–10 or LR− value of 0.1–0.5 is likely to be “useful test”. By contrast, an LR+ value of <2 and an LR− value of >0.5 indicate a “rarely useful test” [29,30].

Cicchetti (1994) provides commonly cited guidelines to distinguish levels that are clinically meaningful as follows: less than 0.70 is unacceptable; between 0.7 and 0.79 is fair; between 0.8 and 0.89 is good; and when it is 0.9 or above, the level of clinical significance is excellent [31].

Here, the instrument’s psychometric properties of Taiwanese version of Internet gaming disorder-short form Likert scale with Likert (IGD-SF-T-L), we used confirmatory factor analysis (CFA) to conducted for validity, the Cronbach’s α for reliability of IGD-SF-T-L, and the ROC (receiver operating curves) to propose the cut-off point for assessing the instrument’s psychometric properties and its corresponding indices for the diagnostic accuracy. The reliability and factor analyses were performed using SPSS software version 24 (SPSS for Windows, SPSS Inc., Chicago, IL, USA). The one-factor confirmatory factor analysis (CFA) was performed by using AMOS version 24 (SPSS for Windows, SPSS Inc., Chicago, IL, USA). Comparisons of the diagnostic accuracy or equivalently the areas under the receiver operating curves (AUCs) were assessed using STATA/SE V13.0 (Stata Corporation, College Station, TX, USA). All statistical tests were two-tailed, and *p* < 0.05 was considered statistically significant.

## 3. Results

We used the CIA (56/57) for defining Internet gaming disorder (IGD). The results of the comparison of the socio-demographic information of the subjects with and without an IGD were summarized in Table 1. The children and adolescents with an IGD problem were older in their age, had parents with a higher average age, were more inattentive, and had more emotional problems than patients without IGD. In addition, patients with IGD: (1) spent more time in online chatting or gaming daily and during weekends (*p* < 0.001), (2) had poorer interpersonal relationships (*p* = 0.002), (3) had higher chance of comorbid diagnosis of oppositional defiant disorder (ODD, *p* = 0.034), and disruptive mood dysregulation disorder (DMDD, *p* = 0.006) than those without IGD. The psychometric property of the proposed IGD-SF-T as follows:

### 3.1. The Reliability Analyses

The factor scores determinacy coefficient, the Cronbach’s α of the Taiwanese version of the IGDS was 0.918, indicating an excellent degree of internal consistency. The Kaiser–Meyer–Olkin value was 0.917, indicating a good sampling adequacy. The dimensionality of the scale was checked by using factor analyses as follows: The results of factor analyses showed that the eigenvalue for the first factor was substantially greater than that for the second factor (5.465 vs. 0.802). In addition, the first factor accounted for 60.72% of the total variance, suggesting that the scale items were unidimensional.

### 3.2. Test the Construct Validity

The one-factor Confirmatory factor analysis (CFA) was applied to test the construct validity of IGD-SF-T-L, as we mentioned in statistical analysis. The good models have root mean square error of approximation (RMSEA) values less than 0.08, our RMSEA = 0.036 implied good models fit. The comparative fit index (CFI) greater than 0.90 is considered to be indicators of good fitting models, our comparative fit index (CFI) = 0.994 still implies good fitting modes. We also applied several other fit indices that were selected to test model fit and still showed good model fit: goodness of fit index (GFI) = 0.95, TLI = 0.991, normed fit index (NFI) = 0.951, parsimony adjustment to the NFI (PNFI) = 0.634, and standardized root mean square residual (SRMR) = 0.032. The fit indices of change in chi-square given the degrees of freedom values were acceptable: χ^2^ = 27.135, *p* = 0.298. Moreover, all nine indicator variables were reliable and valid measures of the latent variable of the IGDS and showed the moderate-to-high factor loadings (β = 0.599–0.831, all *p* < 0.001). As shown in Figure 1, the indices revealed a better model fit of IGDS-SF-T.

### 3.3. Diagnostic Accuracy Indices and Receiver Operating Characteristic (ROC) Curve Analysis

We used the CIA (56/57) as the state variable for the IGD and the Taiwanese version of the IGDS as the test variable. The results of diagnostic accuracy indices, named sensitivity, specificity, LR+ (likelihood ratio), LR−, and AUC (area under curve), with the corresponding selected cut-off values are shown in Table 2. The corresponding indices of accuracy: sensitivity, specificity, LR+, LR−, and AUC were 0.893, 0.826, 5.134, 0.130, and 0.859, respectively. AUC increased to at least 91.8% for determining empirical cut-off point for determining a dichotomized level between the disordered and non-disordered gamers showed that all nine indicator variables were reliable and valid measures of the latent variables of the IGDS (β = 0.599–0.831, all *p* < 0.001) with the moderate-to-high factor loadings.

Using the IGDS-SF-T as the test variable and CIAS > 56 as the state variable, the results of the corresponding receiver operating characteristic (ROC) curve analysis for assessing the discriminate ability of IGD-SF-T were shown in Figure 2 and the area under the ROC curve was 0.918. AUC increased to at least 90% for selected cut-off points, we chose a threshold for the IGD-SF-T that maximized the values of the AUC [32]. As shown in Table 2, the cut-off point proposed for IGD-SF-T-L to indicate a diagnosis of IGD was greater or equal to 10 and denoted by (9/10). That is, we diagnosed an ADHD child or adolescent with IGD if his/her score on the IGDS-SF-T-L was higher than or equal to 10.

In summary, these results indicate that the IGD-SF-T-L is internally consistent and has good structural and criterion validity for this study sample. Accordingly, this new Likert scale diagnostic tool is reliable and valid for diagnosing IGD in children and adolescents with ADHD.

## 4. Discussion

The aim of the study was to exam the psychometric properties of nine-item Internet Gaming Disorder Scale—Short-Form (IGD-SF), a Taiwanese version using Likert Scale. This Likert scale for checking Internet gaming disorder is first novel contribution in this psychometric property of Internet gaming disorder. All values showed Taiwanese version of the DSM-5’s GDS-SF-T Likert scale was considered good psychometric property with high internal consistency as assessed by Cronbach’s α. The construct validity using one factor CFA showed all the indices is generally considered a good fit in and the model fit of researching the data all being very well. All nine indicator variables were reliable and valid measures with the latent variables of the IGDS (β = 0.599–0.831, all *p* < 0.001) being the moderate-to-high factor loadings. The cut-off point proposed for IGD-SF-T Likert scale to indicate a diagnosis of IGD was greater or equal to 10.

In light of the potential new findings, the IGD-SF-T Likert scale satisfies the need for a standardized and psychometrically appropriate tool for assessing Internet gaming disorder among ADHD children and adolescents through the IGD criteria outlined in the DSM-5. Our results corroborate those reported good psychometric property in other studies [12,33] and in line with validity studies of the IGD criteria of the DSM-5 conducted among adolescents with IGD in Slovenia [11].

Worthy to pay attention from this study result are the ADHD youth who were more pathologically video-gamers, with more withdrawal tendencies, more loss of control, and more conflictual tendencies than those ADHD youth without a gaming disorder as previous study found [34]. In addition, these ADHD youth with IGD, here, were indeed found to have a greater number of other comorbid diagnoses of oppositional defiant disorder (ODD) or disruptive mood dysregulation disorder than those without a gaming disorder. The characteristics of these Internet addicted children and adolescents were that they were older in age, had more severe inattentive symptoms, had more emotional problems, had more disruptive mood dysregulation disorders, and had parents with a higher average age. In addition, they spent more time in online chatting or gaming daily and during weekends; moreover, they had poorer interpersonal relationships. The related expertise on caring for ADHD children in Taiwan should pay more attention to these gaming addicted youth.

The limitation of this study was that the study used a convenience sample of children or adolescents with ADHD from the outpatient department of a general hospital. In addition, the data were self-reported, and the results may have associated biases, such as social desirability biases and short-term recall biases among the parents regarding their children. Indeed, the self-evaluated “bias” that might exist, or be due to the scope of the sample, might diminish the research result. Therefore, we recommend that this simple diagnostic tool can be used in the future for other general populations for diagnosing the IGD, in Taiwan or China, with large sample size.

In summary, we suggest that especially for adolescents with ADHD, gaming disorder maybe a “persistent modern umbrella”, meaning modern youth might constantly live under tremendous pressure in the digital using world, which may lead to youth being in a more mentally disordered state. A unified and emergent approach to assessment and treatment of pathological gaming users is needed. This preliminary study will pave the way for further large-scale studies on the diagnoses of IGD among children and adolescents with ADHD in the Chinese general population. In addition to previously highly studied ADHD comorbidity problems of IGD, family attitudes and parental mental health states in overly used Internet-families should be studied. More attention should be paid toward recent ADHD youth with Internet gaming disorders and their families.

## Figures and Tables

**Figure 1 ijerph-18-01492-f001:**
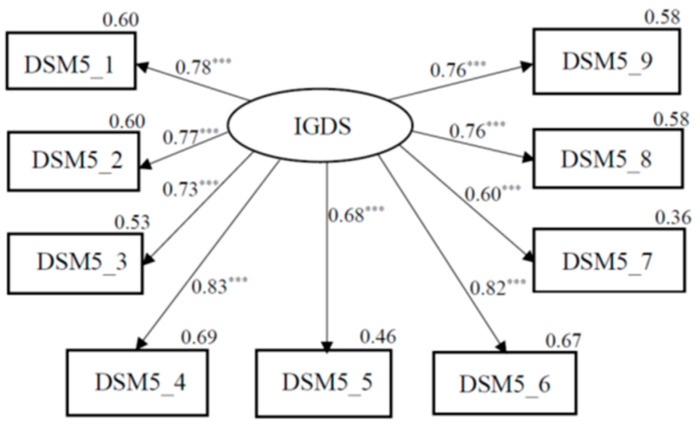
One-factor confirmatory factor analysis (CFA). Circles represent unobserved latent variables. Rectangles represent observed measured variables. Values are standardized path coefficients. The squared multiple correlation (R^2^) value for the dependent variable appears above its rectangle. *** *p* < 0.001.

**Figure 2 ijerph-18-01492-f002:**
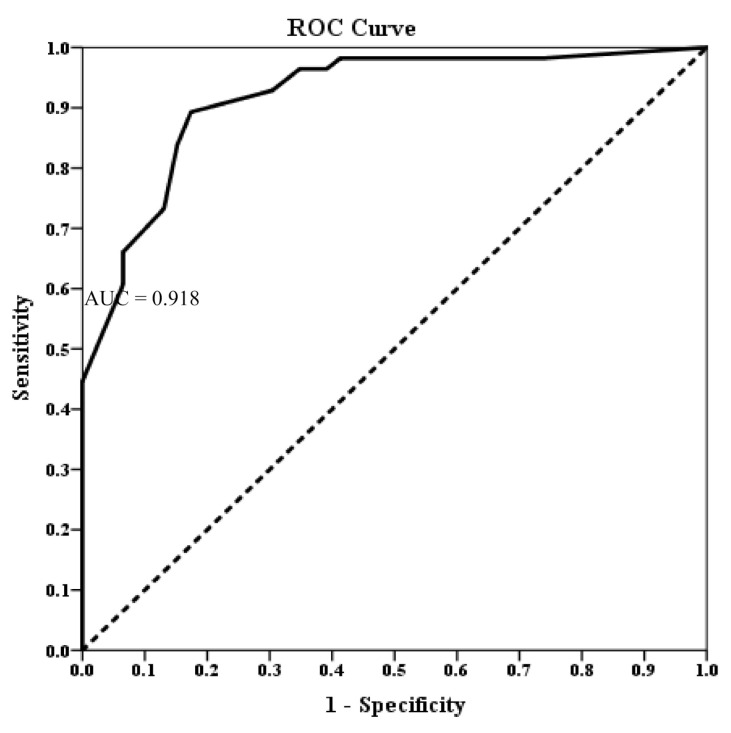
The receiver operating characteristic (ROC) curve.

**Table 1 ijerph-18-01492-t001:** The results of comparing the socio-demographic information of the current sample between Internet Addiction (IA) and non-IA.

Research Question Composition	Demographic Characteristics	IA Internet Addiction (CIAS > 56)		Significance
		No (n = 46)	Yes (n = 56)	*p*-Value
Sex	Male	36 (78.3%)	34 (60.7%)	0.086 ^a^
	Female	10 (21.7%)	22 (39.3%)	
Performance	Middle	24 (53.3%)	23 (41.8%)	0.315 ^a^
	Worse	21 (46.7%)	32 (58.2%)	
Interpersonal	Good	35 (77.8%)	26 (47.3%)	0.002 ^a^
relationship	Bad	10 (22.2%)	29 (52.7%)	
ODD	No	15 (32.6%)	8 (14.3%)	0.034 ^a^
	Yes	31 (67.4%)	48 (85.7%)	
DMDD	No	22 (47.8%)	12 (21.4%)	0.006 ^a^
	Yes	24 (52.2%)	44 (78.6%)	
Comorbidity	Yes	36 (78.3%)	56 (100.0%)	<0.001 ^a^
	No	10 (21.7%)	0 (0.0%)	
Subtype	Combined	33 (71.7%)	32 (57.1%)	0.212 ^a^
	Inattentive	13 (28.3%)	24 (42.9%)	
Family Psychiatric	Yes	9 (19.6%)	12 (21.4%)	1.000 ^a^
History	No	37 (80.4%)	44 (78.6%)	
Sibling with ADHD	Yes	11 (23.9%)	9 (16.1%)	0.331 ^a^
	No	35 (76.1%)	47 (83.9%)	
Daily on line	More than 1h	20 (43.5%)	46 (82.1%)	<0.001 ^a^
Chatting or Gaming	Less than 1h	26 (56.5%)	10 (17.9%)	
Weekend on line	More than 3h	18 (39.1%)	48 (85.7%)	<0.001 ^a^
Chatting or Gaming	Less than 3h	28 (60.9%)	8 (14.3%)	
Treatment Effect	Good	12 (48.0%)	13 (34.2%)	0.303 ^a^
	Bad	13 (520.0%)	25 (65.8%)	
Attend Parent Group	Yes	6 (22.2%)	9 (20.9%)	1.000 ^a^
Program	No	21 (77.8%)	34 (79.1%)	
Compliance	Good	11 (45.8%)	12 (30.8%)	0.414 ^a^
	Bad	13 (54.2%)	27 (69.2%)	
Height		137.98 ± 18.02	149.09 ± 18.55	0.003 ^b^
Weight		34.34 ± 13.60	46.59 ± 18.39	<0.001 ^b^
Age		10.07 ± 3.06	12.25 ± 3.64	0.002 ^b^
Father’s Age		42.67 ± 6.38	46.50 ± 7.81	0.009 ^b^
Mother’s Age		40.20 ± 7.41	43.38 ± 6.88	0.027 ^b^
SNAP_1_9	(Inattention)	19.80 ± 3.06	21.30 ± 3.74	0.031 ^b^
SNAP_10_18	(Hyperactivity)	14.11 ± 6.91	13.89 ± 7.16	0.877 ^b^
SNAP_19_26	(Emotionality)	11.85 ± 6.20	14.18 ± 4.70	0.033 ^b^
DMDD Total		1.09 ± 1.13	1.93 ± 1.04	<0.001 ^b^
CIAS		41.02 ± 10.26	72.52 ± 11.00	<0.001 ^b^
DSMS-SF-T		5.26 ± 4.54	14.75 ± 5.23	<0.001 ^b^

^a^: Fisher’s Exact test; ^b^: Independent *t*-test; ODD: oppositional defiant disorder; DMDD: disruptive mood dysregulation disorder; ADHD: attention deficit hyperactivity disorder; CIAS: Chen’s Internet addiction scale; DSMS-SF-T: the Taiwanese version of the IGDS.

**Table 2 ijerph-18-01492-t002:** The results of diagnostic accuracy indices and the corresponding cut-off values of IGDS-SF-T.

*Cut-off Value* ^a^	(14/15)	(13/14)	(12/13)	(11/12)	(10/11)	(9/10)	(8/9)	(7/8)	(6/7)	(5/6)	(4/5)
Sensitivity	0.446	0.607	0.661	0.732	0.839	0.893	0.929	0.964	0.964	0.982	0.982
Specificity	1.000	0.935	0.935	0.870	0.848	0.826	0.696	0.652	0.609	0.587	0.500
LR+ ^b^	NA	9.310	10.131	5.613	5.515	5.134	3.051	2.772	2.464	2.378	1.964
LR− ^c^	0.554	0.420	0.363	0.308	0.190	0.130	0.103	0.055	0.059	0.030	0.036
AUC ^d^	0.723	0.771	0.798	0.801	0.844	0.859	0.812	0.808	0.786	0.785	0.741

^a^: Cut-off value (14/15): IGDS-SF-T ≥ 15 is IGD; Taiwan IGDS-SF-T ≤ 14 is Non-IGD. ^b^: Likelihood Ratio Positive (LR+ higher than about 5 can be useful in ruling in a disease). ^c^: Likelihood Ratio Negative (LR− values below about 0.2 are useful in ruling out a disease). ^d^: AUC: Area under the receiver operating characteristic (ROC) curve.

## Data Availability

Not report any data.

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
