# Peer review of "Proposing and Validating the Diagnosis Scale for Internet Gaming Disorder in Taiwanese ADHD Adolescents: Likert Scale Method Based on the DSM-5"

_ijerph, 2021, doi:10.3390/ijerph18041492_

Round 1
Reviewer 1 Report
The paper focusses on an important topic in modern society. The treatment of gaming addiction and the related harming impact on persons that comes along with it is a challenge that needs scientific attention. Therefore, the topic picked is most relevant.
Even though, the paper is scientifically sound and backed up by quantitative data, the current state of research on the Likert scale should be presented to a greater extent. The self-evaluated "bias" in the findings, due to the scope of the sample unfortunately diminishes the research conducted.
Furthermore, the conclussion and findings should be more specific to have a greater impact on readers. The "persistent modern umbrella" should be defined and described in greater detail.
Moreover, the language should be double checked in regard to grammar and style.
Author Response
Response to Reviewer 1 Comments
Point 1: The paper focusses on an important topic in modern society. The treatment of gaming addiction and the related harming impact on persons that comes along with it is a challenge that needs scientific attention. Therefore, the topic picked is most relevant.
Response 1: Thank you for your positive reward.
Point 2: Even though, the paper is scientifically sound and backed up by quantitative data, the current state of research on the Likert scale should be presented to a greater extent. The self-evaluated "bias" in the findings, due to the scope of the sample unfortunately diminishes the research conducted.
Response 2: Thank you for your precious remind again. Indeed, the self-evaluated "bias" might exist or due to the scope of the sample might diminishes the research result. Therefore, we have added this comment on the limitation part.(please check the line 393-5 , page 10 )
Point 3: Furthermore, the conclussion and findings should be more specific to have a greater impact on readers. The "persistent modern umbrella" should be defined and described in greater detail.
Response 3: The "persistent modern umbrella" meaning modern youth might constantly live under tremendous digital using world and finally bring youth to be more mentally disordered state. We have added this definition in conclusion part. (please check the line 400-2 , page 11 )
Point 4: Moreover, the language should be double checked in regard to grammar and style.
Response 4: Thank you for kind remind. We have double checked the grammar and style.
Reviewer 2 Report
I will start by stating that, in my view, this paper has some very strong points:
- Its organization is very good,
- the tables and figures are meaningful and interesting and
- the whole presentation helps the reader to understand and appreciate the aim of the paper.
On the other hand, and this is something that I think I believe can be easily addressed in a minor revision, perhaps the authors should provide a more detailed recent literature review focusing on papers that are closely related to the paper in question.
Another remark that I would like to mention is about the contribution and novelty of this paper. Am I to understand that this is done for the first time? In any case, I think it should be made clear and the authors should explain what this work achieves for the first time or what it does better in relation to previous similar works. In other words, they should explain why it deserves to be published as a research article.
Hence, my suggestion to the authors would be to enrich their manuscript with works closely related to this one, and clearly explain the novelty and contribution in comparison to the present state of the art in the literature.
Their command of the English language, although acceptable, still could be improved in order to eliminate a few minor typos.
In conclusion, this work is quite interesting, adequately written, illustrated and presented, and I believe that this paper deserves publication after minor revisions.
Author Response
Response to Reviewer 2 Comments
Point 1: I will start by stating that, in my view, this paper has some very strong points:
Its organization is very good, the tables and figures are meaningful and interesting and
the whole presentation helps the reader to understand and appreciate the aim of the paper.
Response 1: Thank you for your positive feedback.
Point 2: On the other hand, and this is something that I think I believe can be easily addressed in a minor revision, perhaps the authors should provide a more detailed recent literature review focusing on papers that are closely related to the paper in question.
Response 2: Thank you for your good remind again. We had found one recent literature review focused on this topic. Therefore, we have added this literature review on introduction part as following: “According to a comprehensive review, there is a close association between gaming disorder and ADHD (Gonzalez-Bueso et al. 2018)”.(please check ine 98-9 , page 3 )
Point 3: Another remark that I would like to mention is about the contribution and novelty of this paper. Am I to understand that this is done for the first time? In any case, I think it should be made clear and the authors should explain what this work achieves for the first time or what it does better in relation to previous similar works. In other words, they should explain why it deserves to be published as a research article.
Response 3: “This Likert scale for checking internet gaming disorder is first novel contribution in this psychometric property of internet gaming disorder.” We have added this point inside the conclusion part. ( please check line 349-51 , page 10 )
Here the instrument’s psychometric properties of Taiwanese version of Internet gaming disorder-short form Likert scale with Likert (IGD-SF-T-L), we used confirmatory factor analysis (CFA) to conducted for validity, the Cronbach’s α for reliability of IGD-SF-T-L, The ROC (receiver operating curves) to propose the cut-off point for assessing the instrument’s psychometric properties and its corresponding indices for the diagnostic accuracy. We have added this point inside the end of method part. (please check line 226-33, page 5 )
Point 4: Hence, my suggestion to the authors would be to enrich their manuscript with works closely related to this one, and clearly explain the novelty and contribution in comparison to the present state of the art in the literature.
Response 4: Thank you for such precious comment to enrich this article.
Point 5: Their command of the English language, although acceptable, still could be improved in order to eliminate a few minor typos.
Response 5: Thank you for kind remind. We have double checked the grammar and style.
Point 6: In conclusion, this work is quite interesting, adequately written, illustrated and presented, and I believe that this paper deserves publication after minor revisions.
Response 6: Thank you for giving us a chance to start minor revision. We wish this article could be accepted and published in your honourable journal.
Reviewer 3 Report
I have some comments for the authors’ consideration.
The major issue of this manuscript is language quality, which should be improved before next round of review.
How did you recruit the study sample? Please indicate this clearly.
For the method, did you have any approach to improve the quality of data collection. Doing a validation study is of great importance to control data quality. If you had, please clarify in the method section.
The table in this study should be modified as current tables were not well-readable.
In the discussion, I suggest you should compare your research findings with previous studies, which could be a better approach to understand your results in a comprehensive way.
Author Response
Response to Reviewer 3 Comments
Point 1: I have some comments for the authors’ consideration.
The major issue of this manuscript is language quality, which should be improved before next round of review.
Response 1: Thank you for your kind remind. We have improved the language quality.
Point 2: How did you recruit the study sample? Please indicate this clearly.
Response 2: Thank you for your good remind again. We had shown the way we recruit patient as following content inside the method part.
“A total of 102 children and adolescents (mean age = 11.16 ± 3.35 years, 68.6 % boys) were recruited from the outpatient units of the Mackay Memorial Hospital (MMH) in Taipei, Taiwan. The MMH Institutional Review Board (IRB) approved the research protocol. Written informed consent was obtained from each subject as per the IRB guidelines. According to the inclusion criteria, boys and girls with ADHD aged 7–18 years were enrolled. If the patients or their parent(s) or caregiver(s) suspected they had psychotic disease, mental retardation, or other mental conditions that could prevent them from completing the study, they were excluded. After obtaining signed consent from a legal guardian, each participate were interviewed for the following measurements. “
The MMH Institutional Review Board (IRB) approved the research protocol, no. is 19MMHIS387e. Written informed consent was obtained from each subject as the IRB guidelines.
Point 3: For the method, did you have any approach to improve the quality of data collection. Doing a validation study is of great importance to control data quality. If you had, please clarify in the method section.
Response 3: Beside this stud is approved by IRB and follow the IRB guideline. Also, on the instrument’s psychometric properties, we used Confirmatory factor analysis (CFA) to conducted for validity, the Cronbach’s α for reliability of IGD-SF-T-L, The ROC (receiver operating curves) to propose the cut-off point for assessing the instrument’s psychometric properties and its corresponding indices for the diagnostic accuracy.
But for the clarity of this article, we have added this short summary inside the method part as following:
“Here the instrument’s psychometric properties of Taiwanese version of Internet gaming disorder-short form Likert scale with Likert (IGD-SF-T-L), we used Confirmatory factor analysis (CFA) to conducted for validity, the Cronbach’s α for reliability of IGD-SF-T-L, The ROC (receiver operating curves) to propose the cut-off point for assessing the instrument’s psychometric properties and its corresponding indices for the diagnostic accuracy. (please check line 226-33 , page 5)
Point 4: The table in this study should be modified as current tables were not well-readable.
Response 4: Thank you for such precious comment. We have modified the table become readable.
Point 5: In the discussion, I suggest you should compare your research findings with previous studies, which could be a better approach to understand your results in a comprehensive way.
Response 5: Thank you for kind remind. We have double checked the grammar and style.